# Adolescent COVID-19 Vaccine Decision-Making among Parents in Southern California

**DOI:** 10.3390/ijerph19074212

**Published:** 2022-04-01

**Authors:** Suellen Hopfer, Emilia J. Fields, Magdalen Ramirez, Sorina Neang Long, Heather C. Huszti, Adrijana Gombosev, Bernadette Boden-Albala, Dara H. Sorkin, Dan M. Cooper

**Affiliations:** 1Department of Health, Society & Behavior, Program in Public Health, School of Population and Public Health, University of California, Irvine, CA 92697, USA; fieldse@uci.edu (E.J.F.); magdalr2@uci.edu (M.R.); snlong@uci.edu (S.N.L.); bbodenal@hs.uci.edu (B.B.-A.); 2Pediatric Psychology, Children’s Hospital of Orange County (CHOC), Orange, CA 92686, USA; hhuszti@choc.org; 3Institute for Clinical and Translational Science, University of California, Irvine, CA 92697, USA; agombose@hs.uci.edu (A.G.); dsorkin@hs.uci.edu (D.H.S.); dcooper@hs.uci.edu (D.M.C.); 4Department of Epidemiology and Biostatics, University of California, Irvine, CA 92697, USA; 5Department of Neurology, School of Medicine, University of California, Irvine, CA 92697, USA; 6Department of Medicine, Institute for Clinical and Translational Science, University of California, Irvine, CA 92697, USA; 7Department of Pediatrics, Institute for Clinical and Translational Science, University of California, Irvine, CA 92697, USA

**Keywords:** adolescent COVID-19 vaccination, parent vaccine decision-making, 5C vaccine hesitancy model, vaccine acceptance, vaccine confidence, public health vaccine communication

## Abstract

Adolescent COVID-19 vaccination has stalled at 53% in the United States. Vaccinating adolescents remains critical to preventing the continued transmission of COVID-19, the emergence of variants, and rare but serious disease in children, and it is the best preventive measure available to return to in-person schooling. We investigated parent–adolescent COVID-19 vaccine decision-making. Between 24 February and 15 March 2021, we conducted surveys and 12 focus groups with 46 parent–adolescent dyads in Southern California. Parents and adolescents completed a survey prior to participation in a focus group discussion, which focused on exploring COVID-19 vaccine acceptance or uncertainty and was guided by the 5C vaccine hesitancy model. Parents uncertain about vaccinating adolescents expressed low vaccine confidence and high COVID-19 disease risk complacency. Parents who accepted COVID-19 vaccination for adolescents expressed high confidence in health authority vaccine recommendations, high perceived COVID-19 risk, and collective responsibility to vaccinate children. Additionally, unique pandemic-related factors of vaccine acceptance included vaccinating for emotional health, resuming social activities, and vaccine mandates. Among parents, 46% were willing to vaccinate their adolescent, 11% were not, and 43% were unsure. Among adolescents, 63% were willing to vaccinate. Despite vaccine availability, 47% of adolescents remain unvaccinated against COVID-19. Factors associated with vaccine uncertainty and acceptability inform health care practitioner, school, community, and public health messaging to reach parents and adolescents.

## 1. Introduction

An estimated 72.8 million children live in the United States, comprising 22.2% of the population. Adolescents aged 12–17 comprise 7.7% of that group, or 25.1 million children [1,2]. Reaching community immunity against COVID-19 and preventing the continued spread of COVID-19 and emergence of new variants requires pediatric vaccination [3]. Adolescents are vectors of SARS-CoV-2 transmission [4,5] and therefore, each adolescent vaccinated reduces the risk of viral infection in the community [6,7]. These are indirect benefits of vaccination for adolescents. Nevertheless, these indirect benefits are justified, given vaccination’s role in reducing the continued spread of emerging variants such as the delta or omicron variant, offering greater protection for children attending in-person school and social activities, and preventing disruptions to daily life [4].

However, vaccinating adolescents against COVID-19 also has direct benefits. Although severe illness in adolescents is rare, vaccination is warranted, given that it can effectively prevent severe illness, hospitalization, and long COVID-19 [8,9]. Since the beginning of the pandemic, more than 7.5 million children including adolescents have been confirmed positive for COVID-19, representing 1 in 10 children or an estimated 20.8% of cumulative COVID-19 cases, as of December 2021 [10]. Only 24 states report on hospitalization among children, which appears to be uncommon [10,11,12]. However, more than 8000 children have experienced severe illness, and among adolescent COVID-19-related hospitalizations, there have been 721 reported deaths, many of which occurred among Black and Hispanic adolescents [13]. Vaccines are available and provide protection against severe COVID-19, chronic post-COVID-19 sequelae (i.e., long COVID-19), and rare albeit serious complications, such as multisystem inflammatory disease in children [14,15]. Hospitalization among unvaccinated adolescents is 10 times higher than vaccinated adolescents [16].

Southern California, and Orange County in particular, features diverse communities in terms of racial and ethnic makeup, political ideologies, and a history of communities of color being disproportionately affected by COVID-19 during the pandemic [17,18,19]. Since the 2000s, Orange County has witnessed dramatic growth, with more than 3 million residents, of which 35% are Hispanic or Latino and 26% of those 35% who speak Spanish at home; 21% of its population is Asian; 46% of households speak another language at home; an estimated 35% lives in multigenerational households; and 30% was born in another country [18]. As part of California’s COVID-19 policy, Orange County families also experienced being part of the first state in the country to close in-person schools and transition to virtual learning early in the pandemic. By early 2021, many families were eager for their children to return to school, and families, pediatricians, epidemiologists, and scholars were anticipating the emergency approval and recommendation for the COVID-19 vaccine to become available to adolescents [20,21].

On 10 May 2021, the Pfizer BioNTech COVID-19 vaccine became available to adolescents aged 12–17 in the United States after the U.S. Food and Drug Administration issued an emergency use authorization [22]. Although there was an initial peak in vaccine uptake in June 2021, 7 months later, at the end of 2021, only 53% of adolescents had completed their vaccination series [13]. Parental willingness to vaccinate adolescents against COVID-19 varies widely [21,23,24]. Research exploring parental decision-making is needed to inform the design of effective communication that promotes vaccination. Rare incidents of vaccine-induced myocarditis [25,26] also affected parent decision-making, especially among parents of adolescent sons; however, the Advisory Committee on Immunization Practices and the Centers for Disease Control and Prevention (CDC) deemed that the vaccine benefits outweighed the rare myocarditis risk for vaccinating adolescents, especially adolescents with comorbidities [26].

The 5C vaccine hesitancy model gives insight into factors that contribute to vaccine uncertainty or acceptance. The 5C model consists of five constructs: (a) vaccine confidence (trust in the safety and effectiveness of vaccines, the development process, the system that delivers them, and the competence of health care providers); (b) risk complacency (perceptions regarding disease risk and vaccine utility); (c) vaccine constraints (practical barriers to vaccinating such as availability, affordability, and health literacy); (d) vaccine calculation (the need for extensive information searching and weighing vaccine benefits and risks); and (e) collective responsibility to vaccinate for others [27]. The following study was guided by the 5C vaccine hesitancy model and aimed to qualitatively explore parental COVID-19 vaccine decision-making for their adolescent children.

## 2. Materials and Methods

### 2.1. Study Design, Participants, and Setting

The study consisted of recruiting parent–adolescent dyads from Orange County, California, to explore factors that contribute to COVID-19 vaccination uncertainty or acceptance. People of color became a majority minority in the county in the early 2000s. Orange County has outpaced the nation in its dramatic population growth and demographic transformation, driven by growing Latino and Asian American populations. The 35% Hispanic minority, with 26.8% speaking Spanish at home and many of whom live in Santa Ana and Anaheim, have been disproportionately affected by COVID-19 [17]. Another sizeable minority in Orange County is Asians at 21% of the population. A sizable subgroup and enclave of Vietnamese people reside in the county and are characterized by diverse political ideologies. More than 700,000 people in California live in multigenerational households, with a sizeable portion in Orange County, and 30% of county residents were born in another country [18]. Our recruitment sampling methods mirrored the county’s demographics, sampling based on the diverse race and ethnic makeup, Spanish-speaking residents, multigenerational households, diverse occupations, educational attainment, and income.

Parents and adolescents completed a one-time electronic survey prior to participation in a one-time virtual focus group discussion. The 5C model guided development of the survey and the focus group discussion guide. Analysis of parent COVID-19 vaccine decision-making for their adolescent children is presented in this article, as are the adolescent survey results. Dyadic analysis of adolescent–parent discussion is reported separately.

### 2.2. Data Collection

Between 24 February and 15 March 2021 (prior to the emergency authorization of the Pfizer BioNTech vaccine for adolescents aged 12–17), we conducted 12 virtual focus groups with parent–adolescent dyads in Orange County. Interested parents contacted the study coordinator, with whom eligibility was confirmed per telephone. Families, i.e., parent–adolescent dyads, provided verbal informed consent by telephone before they agreed to participate in the study. The study purpose and involvement was explained (one-time survey and one-time virtual focus group discussion), as was the voluntary nature of participation and the USD $50 compensation. Participants received a study information sheet (written summary of what was reviewed) by email, and parents also received a link for each parent and adolescent to an electronic survey to complete prior to the virtual focus group. In two instances, the email was sent to the adolescent when the parent did not have an email address. A 24-h period was given prior to scheduling focus groups. Of the 12 focus groups, nine were held in English and three in Spanish. Each focus group was moderated by one of five trained moderators. After logging into the Zoom meeting together, parents and adolescents introduced themselves. Parents were then asked to step away while adolescents had a 30-min discussion with the moderator. Parents were then asked to return (and adolescents to step away) to participate in a 60-min discussion. Focus groups were video recorded for accuracy, and participants were asked to keep their web cameras on. The size of the focus groups ranged from three to six participants. Six focus groups were composed of three parents, three groups had four parents, two groups had five parents, and one group had six parents. Households received a USD $50 electronic gift card as compensation.

### 2.3. Recruitment

We virtually recruited with the assistance of community partners who distributed electronic flyers (in English and Spanish) through local schools, listservs, and Children’s Hospital of Orange County. Parents were recruited from communities known to have higher COVID-19 morbidity and mortality risk attributed to race and ethnicity (e.g., Santa Ana), live in multigenerational households, have children with chronic conditions, and reflect a range of parental occupations [17]. Parents were screened by telephone for eligibility: (a) parent of a middle or high school student aged 11–18 years old; (b) English or Spanish speaker; and (c) Orange County resident. Parents with multiple eligible adolescents were asked to consider one for enrollment in the study. The study coordinator verbally reviewed the study purpose and participant involvement, as previously described, then the parent and the adolescent agreed to participate verbally by telephone prior to scheduling the focus group (24 h were given prior to scheduling). Families received the study information by email to the parent. The study was approved by the university’s institutional review board to comply with human protections (HS#2021-6423). Signed informed consent was waived.

### 2.4. Parent Survey and Focus Group Discussion Guide

The electronic 54-item survey and semistructured focus group guide were developed for parents and adolescents (see Appendix A). This article reports predominantly on the parents’ responses, focusing on the discovery of factors contributing to vaccine uncertainty and confidence. Both study instruments (survey and focus group discussion guide) were informed by the 5C vaccine hesitancy model to investigate aspects of vaccine confidence, COVID-19 perceived risk for adolescents, constraints to vaccinating, perceptions of collective responsibility to vaccinate, and calculation (extent of deliberation about vaccine benefits and risks). Survey domains included questions regarding parent–child vaccine communication, ranking of COVID-19 vaccine concerns and motivators to vaccinate, parenting style, positive expectancies regarding the pending approval of adolescent COVID-19 vaccination, decision-making under conditions of uncertainty, vaccine information sources, and sociodemographics, including adolescents’ receipt of the influenza vaccine in the last year. See Table 1 for parent survey question domains. The adolescent survey and focus group discussion guide were similar but adjusted for an adolescent perspective and language.

The focus group discussion guide consisted of five parts: (a) parental decision-making for vaccinating their children in general, (b) parental decision-making for COVID-19 vaccination, (c) desired COVID-19 vaccine information related to adolescents, (d) vaccine information sources, and (e) thoughts on vaccinating children 11 years old or younger (see Table 2).

### 2.5. Data Analysis

For the survey, the data analysis consisted of calculating descriptive statistics (i.e., frequencies, percentages, and means) for survey responses related to the COVID-19 vaccination status of parents, vaccine intent for children, adolescent influenza vaccination status, ranking of COVID-19 vaccine concerns and motivators, and family (parent and adolescent) sociodemographic information.

Qualitative data analysis was conducted of the transcribed audio-recordings of the parent and adolescent focus group discussions. Data were transcribed verbatim for accuracy and personal identifiers were replaced with pseudonyms. The Spanish recorded data were transcribed into Spanish first, then back translated into English to ensure accuracy. We took a phronetic iterative approach [28,29,30] to analyze the data and discover emergent themes relevant to parent COVID-19 vaccine decision-making. Phronetic refers to the Greek word phronesis, which prioritizes contextual knowledge. An iterative approach to data analysis involved first inductively analyzing the data line-by-line and tagging segments with descriptive codes (describing what was said and developing labels), and then using the 5C vaccine hesitancy model to deductively analyze the extent to which parents expressed aspects of the five Cs of vaccine acceptability: confidence, COVID-19 risk complacency, constraints (practical barriers), calculation (weighing benefits and risks to vaccinating), and collective responsibility to vaccinate children against COVID-19 once the emergency authorization for adolescents is approved [28,30]. All coders initially independently read the transcript data (i.e., data immersion). Segments of data were then tagged and labeled with descriptive codes (primary coding). A codebook was then developed, containing transcript quotes exemplifying aspects of research questions (e.g., what influences decision-making regarding vaccinating, with 32 descriptive codes for contributors to vaccine uncertainty and 30 descriptive codes for contributors to vaccine acceptability). For data analysis of the Spanish language data, two coders fluent in Spanish analyzed the data, whereas another coder analyzed the English version of the Spanish data. Coders met to discuss emergent codes from the data and organize codes into themes. Secondary data analysis involved organizing the descriptive codes into higher-order themes as they related to answering research questions regarding what factors contribute to parent acceptance or uncertainty to vaccinate their children against COVID-19 when the vaccine is approved for adolescents. Data and supporting quotes were organized into parent COVID-19 vaccine decision-making themes, which expressed vaccine uncertainty and acceptance, corresponding to the 5C model constructs [27,31]. Vaccine decision-making factors not recognized by the 5C model were identified and characterized as additional, newly discovered unique pandemic-related factors motivating vaccine acceptance or uncertainty.

## 3. Results

The 12 focus groups featured 46 parents and 46 adolescents. The mean parent age was 45 and the mean adolescent age was 14. All parents except one were mothers, most parents (74%) had a bachelor’s degree education or higher, nearly half of parents (48%) reported being Latino, 22% of parents reported having an adolescent with a chronic medical condition, and 30% of parents lived in multigenerational households. More than half of adolescents identified as male, most (93%) attended online distance learning for school at the time of the study, and many (67%) had been vaccinated against influenza in the last year. Nearly half of parents (46%) were willing to vaccinate their adolescent against COVID-19, 11% were not, and 43% were unsure. Many adolescents (63%) were willing to be vaccinated, 20% neither agreed nor disagreed, and 17% said they were not willing to be vaccinated. Among parents, 28% were vaccinated against COVID-19 at the time of the study. Sociodemographics are reported in Table 3.

Survey results of parents’ and adolescents’ ranking of COVID-19 vaccination concerns and motivators are shown in Figure 1, Figure 2, Figure 3 and Figure 4. Their concerns centered on vaccine confidence, with top concerns for parents focusing on long-term vaccine side effects. One of the adolescents’ top concerns in addition to confidence was fear of needles. Parents’ and adolescents’ motivators to vaccinate showed that confidence in the vaccine ranked at the top. Additionally, unique pandemic-related factors that motivated willingness to vaccinate included vaccinating for social benefit and collective responsibility.

For the qualitative analysis, results are reported as constructs from the 5C model that contributed to COVID-19 vaccine uncertainty or acceptance among parents (see Figure 5).

### 3.1. COVID-19 Vaccine Uncertainty

From analysis of parents’ discussions expressing vaccine uncertainty, two of the five Cs emerged as relevant: low confidence in the COVID-19 vaccine and high complacency about disease risk for adolescents. Constraints, calculation, and collective responsibility constructs did not emerge among parent expressing vaccine uncertainty.

#### 3.1.1. Low Confidence in the COVID-19 Vaccine

A key theme regarding parental vaccine uncertainty was expressions of low confidence in the COVID-19 vaccine. Many parents (*n* = 32 of 46) expressed low vaccine confidence in 20 distinct ways. Parents raised concerns about the vaccine’s safety and effectiveness and described mistrust in pharmaceutical companies, regulatory agencies, public health authorities, and health care providers. For vaccine safety, parents expressed concerns over unknown long-term side effects on their adolescent’s health and development, an unpredictable immune response to the vaccine, the possibility of the vaccine exacerbating pre-existing medical condition such as asthma, and uncertainty about vaccine ingredients. Parents who expressed greater worry about vaccine effects expressed concerns for younger (i.e., preteen) and smaller children being able to handle the immune response induced by the vaccine. When discussing vaccine effectiveness, parents expressed concerns over short-lived immunity and a lack of protection against emerging variants.

Parents’ mistrust in pharmaceutical companies and regulatory agencies was expressed in their comments about the COVID-19 vaccine development process being rushed and there being “corners cut”. Parents questioned whether pharmaceutical companies valued and prioritized the health of children. Additionally, some parents expressed that vaccine companies lacked transparency about the vaccine’s ingredients and technologies (i.e., mRNA platform), which led to skepticism and hesitation. Parents’ vaccine uncertainty was expressed in their discussion about the political and public health pressure to approve and roll out a vaccine quickly.

Parents also reported mistrust in public health authorities and health care providers. A few participants doubted public health agencies such as the CDC, given their inconsistent mask-wearing recommendations. Some parents noted a preference for discussing the COVID-19 vaccine with family or friends working in health care, rather than their own provider or their adolescent’s pediatrician. These parents perceived some health care providers as having a vested position in not questioning the credibility of the vaccine. A few parents also expressed low confidence in providers knowing much about the vaccine or its ingredients.

Ultimately, for some parents, vaccine concerns and mistrust undermined the confidence needed to view the vaccination as a protective measure for adolescents. Parents described waiting months to years to vaccinate their adolescent, to observe the vaccine’s effect on other children. Uncertain parents emphasized needing more long-term clinical data and time to consider vaccinating their adolescent.

#### 3.1.2. High Complacency regarding COVID-19 Disease Risk

Slightly more than one third of parents (32.6%, *n* = 15), did not perceive COVID-19 as a significant health threat to their adolescent children or perceived COVID-19 risk as manageable. High disease complacency was expressed in 12 ways. Parents reported that their adolescent either was not at risk of contracting SARS-CoV-2 or had already contracted the virus and the risk was manageable. Parents cited sheltering in place, social distancing, and online distance learning as measures that safeguarded children. Managing risk of infection was also facilitated by a parent’s ability to control their adolescent’s social activity. Parents who reported less compliance with preventive measures often acknowledged their adolescent’s risk of exposure but normalized the risk. One parent discussed not feeling at risk because the virus seemed impossible to contract despite them attending several social functions during the pandemic. Some parents attributed this lack of susceptibility to the adolescent’s immune system.

Low risk perceptions related to disease severity were expressed among parents of healthy children. Low risk perceptions persisted even among parents who had personal experiences with COVID-19 or who knew someone who had contracted the disease. For example, one mother whose son previously had COVID-19 described the illness as “manageable” and said that her son was “resilient”. She described being more worried about the stigma and inconvenience of having an infected child than the disease. Parents of young adolescents did not perceive the vaccine as necessary and reported an intent to delay vaccination due to low risk perceptions. Participant quotes highlighting these findings are presented in Table 4.

### 3.2. COVID-19 Vaccine Acceptability

From analysis of parent discussions expressing vaccine acceptance, three of the five Cs emerged as relevant: high confidence in the COVID-19 vaccine, low complacency regarding disease risk, and a collective responsibility to vaccinate. Three additional unique pandemic-related factors emerged as relevant in shaping parents’ acceptance of vaccinating adolescents: vaccinating for emotional health, resuming social activities, and vaccine mandates as motivators.

#### 3.2.1. High Confidence in the COVID-19 Vaccine

Close to 40% (39.1%, *n* = 18) of parents expressed high vaccine confidence, and they expressed it in seven ways. They discussed trusting their doctor, expressing trust in science, personal observations that others were vaccinated without complications, family norms to vaccinate, perceiving the vaccine as safe due to vaccine trial information, messages from public health authorities encouraging vaccination, and perceiving their older adolescent children as able to handle the immune response like adults. An explicit, strong health care provider recommendation to vaccinate adolescents certainly carried weight in deciding whether to vaccinate adolescents against COVID-19. One parent expressed that she needed to be “talked off the ledge” by her children’s pediatrician to vaccinate her children against COVID-19 when it becomes available. Parents often described health care providers as educated and positioned to increase vaccine literacy. Participants expressed interest in wanting to understand the science behind the vaccine before deciding whether to vaccinate. Parents of adolescents with underlying health conditions valued recommendations by their child’s health care specialist. Parents also stated they would consider the recommendations of family and friends in the health care field. Overall, parents cited trust in science and providers as a reason for being more accepting of a COVID-19 vaccine for adolescents.

Greater confidence in the COVID-19 vaccine was ascribed by parents when they articulated being more confident about vaccinating, because they personally knew individuals who had been vaccinated without side effects or complications. Additionally, parents described having confidence in vaccinating because of growing up in a family or household where vaccinating was perceived as protective and normative.

#### 3.2.2. Low Complacency regarding COVID-19 Disease Risk

Parents who intended to vaccinate their adolescents were more concerned about the risk of disease than vaccine side effects. Low complacency was expressed in eight ways by 37% of parents (*n* = 17). Parents voiced concerns about severe disease and death from COVID-19. Some parents were worried that contracting SARS-CoV-2 would exacerbate their adolescent’s pre-existing health conditions. Parents with high risk perceptions shared stories about prior illnesses in the household and stated that these experiences set the context for why they intended to vaccinate their adolescent.

#### 3.2.3. High Collective Responsibility to Vaccinate

Collective responsibility was expressed in eight ways by 43.5% of parents (*n* = 20). Parents reported wanting to vaccinate their adolescent to protect their social circle: older grandparents living in the household, the adolescent’s teachers, neighbors, local grocery workers, and other individuals with whom their adolescent interacts (e.g., friends, friends’ grandparents, and teammates). Although parents did not perceive children to be at risk of severe disease, many worried about their adolescent being a source of viral transmission. Fear of infecting others was associated with feelings of guilt. Participant quotes highlighting vaccine acceptance are presented in Table 5.

#### 3.2.4. Vaccinating for Emotional Health

Unique to the pandemic, 23.9% of parents (*n* = 11) discussed vaccinating their adolescent for emotional health. Parents discussed how socially isolated their children had been in the past year and expressed the desire for their adolescent to attend school in person. Parents worried about the long-term impact of distance learning on their child’s development. Parents also described the COVID-19 vaccine as a way to end the pandemic and return to normal life, citing vaccine intentions for their own emotional health and describing fatigue from living in fear during the pandemic. Parents perceived vaccination as offering a sense of security and emotional relief.

#### 3.2.5. Vaccinating to Resume Social Activities

Related to emotional health, 21.7% of parents (*n* = 10) discussed vaccinating to resume social activities, including in-person schooling, afterschool activities, sports events, school graduation (from middle or high school), birthday parties, sleepovers, and socializing with family and friends. Vaccinating adolescents would allow them to safely resume social activities.

#### 3.2.6. Vaccinating Due to Mandates

A smaller percentage of parents, 17.4% (*n* = 8), reported that they would vaccinate their adolescents only if vaccines were mandated. Vaccine mandates included those issued by the adolescent’s middle or high school and the state, but parents also discussed of mandates for college enrollment, travel, and work. A few parents expressed concerns over schools preemptively mandating vaccination. Participant quotes highlighting pandemic-related factors are presented in Table 6.

## 4. Discussion

This study identified how parents expressed COVID-19 vaccine acceptance or uncertainty for their adolescent children early in the pandemic, when vaccination for adults had just recently been approved by emergency authorization and approval for adolescent COVID-19 vaccination was still pending (the study was conducted February and March 2021, and the vaccine for adolescents was approved in May 2021). Despite the COVID-19 vaccine being available for adolescents in the United States since May 2021, adolescent vaccination stalled at 53% at the end of 2021 [2], with vaccination rates varying significantly across states and communities. An estimated 9.5 million 12- to 17-year-olds still need to be vaccinated [32]. Even in Orange County, California, adolescent vaccination rates vary widely by city from 49% to 95% as of February 2022 [33]—a seeming microcosm of the nation, whose adolescent vaccination rates also vary widely by state and community [33]. The success of any vaccine program to reduce the burden of COVID-19 depends on high vaccine acceptance and uptake, including adolescents but most importantly, their parents [34]. Vaccinating adolescents plays a critical role in limiting the spread of COVID-19 [6,9] and is the best preventive measure available to offer safe in-person schooling. In addition to the direct benefit of minimizing rare yet possibly severe COVID-19 disease, the indirect benefits of vaccinating adolescents, i.e., for preventing community transmission, prevail and are critical. The CDC released a statement highlighting the need to urgently increase COVID-19 vaccination coverage, particularly in light of the spread of new variants [35].

Understanding parent vaccine decision-making for adolescents is important for delivering tailored vaccine messaging to subgroups of parents who may turn to health care practitioners, public health, community advocates, and school officials for vaccine recommendations or cues about its importance. In our study, parents’ uncertainty about vaccinating their adolescent against COVID-19 reflects vaccine attitudes among parents in Orange County, California, during the early pandemic—a period of high uncertainty. During this time, parents’ vaccine attitudes reflected low vaccine confidence and high complacency about COVID-19 risk in children among parents who were uncertain. Parent vaccine attitudes may be changing among some, given the rising proportion of COVID-19 infections among children (18.6%) and that for subgroups of children, especially unvaccinated children, COVID-19 disease risk may result in hospitalization and in rare cases, multisystem inflammatory disease or death [10].

Among parents confident in vaccinating their adolescents at a time when the vaccine had not received emergency authorization, high vaccine confidence, low risk complacency, and a high collective responsibility to vaccinate characterized vaccine acceptance attitudes. Three additional vaccine acceptance factors were discovered, which have not been recognized in vaccine-specific theories: vaccinating for emotional health, to resume social activities, and in response to mandates. Parents uncertain about vaccinating their children expressed low confidence in the vaccine and mistrust in government regulatory agencies, pharmaceutical companies, and public health officials. Parents’ mistrust exhibited a spillover effect from early pandemic blunders, when public health authorities gave inconsistent messaging on masks [36,37].

Vaccine hesitancy factors recognized by the 5C model that were not raised by parents at the time of the study (February–March 2021) included constraints (not being able to access vaccination) and calculation (extensively searching for COVID-19 information). Similar to a national study [21], low vaccine confidence because of unknown long-term outcomes was expressed by parents, as were questions about vaccine ingredients, vaccine efficacy given variants, and wanting more safety information about clinical trials involving adolescents. The vaccination status of the parent did not necessarily align with the parent’s vaccine attitude to vaccinating their child, nor did the parent’s vaccine attitude always align with their child’s vaccine attitude. That is, some parents were vaccinated but did not want or deem it necessary to vaccinate their children. Additionally, in some cases, parents expressed not intending to vaccinate their adolescent children, whereas the adolescent child expressed the intent or desire to vaccinate. Although preteen children (5th and 6th graders) tended to mirror their parent’s vaccine attitudes, older adolescent children (high school age) in some cases had greater access to independent vaccine information through school or social media and as a result, expressed greater confidence in wanting to vaccinate once the vaccine became available. Among the 28% of vaccinated parents at the time of the study in February–March 2021, one parent did not want her children vaccinated because she had experienced adverse vaccine reactions, whereas another expressed not perceiving the need for her sons to be vaccinated because they had already contracted COVID-19 and she felt the risk was manageable. Achieving community immunity did not factor into this parent’s vaccine attitude regarding her sons.

Parents whose adolescent child had comorbidities were both uncertain about and accepting of vaccination. These parents expressed turning to their physician for guidance. Similar to findings from a national survey [38], our findings suggest the continued importance of having individuals with medical credibility, e.g., pediatricians, public health officials, and school nurses, answer parents’ questions to increase vaccine confidence. Additionally, findings suggest the continued importance of having localized community efforts to increase vaccine education, opportunities to answer parents’ questions, and access to vaccination in the community. In light of the increasing spread of variants in the United States and the rapidly changing risk environment for adolescents [35], COVID-19 vaccine discussions with parents and adolescents continue to be critical. Our findings provide guidance on key areas to emphasize in discussions with parents and their adolescents who have yet to be vaccinated.

Factors associated with vaccine acceptance and uncertainty may inform effective messaging. With the continued spread of COVID-19, parents uncertain about vaccinating need to be reminded by trusted pediatricians of the importance and priority of vaccinating adolescents. We now know that risk to adolescents, especially those with comorbidities, may be higher, especially for potentially severe COVID-19 or long COVID-19. Receiving explicit vaccine recommendations from a personal pediatrician and positive vaccine norms from the community (especially family members) may signal to parents who are uncertain that prioritizing vaccination is important. Provider recommendation was explicitly mentioned as important by parents, in contrast to a national survey study of parents, who indicated that a health care provider vaccine recommendation would not change their vaccine intentions [21]. Lessons from other adolescent vaccine attitude studies (e.g., HPV vaccination) suggest that parents receiving a strong explicit vaccine recommendation from the family pediatrician is a necessary condition to increase vaccination [39,40]. In addition to pediatrician recommendation, community and school vaccine norms may have an influence on parents’ decisions, because some families visit their pediatrician less during the pandemic and may be in more regular communication with schools for COVID-19 guidance. Nationally, adolescent immunizations have dropped during the pandemic [38,41].

Trusted pediatric practitioners and school and community nurses who engage families in vaccine conversations may benefit in shifting the conversation to social benefits or the collective responsibility of vaccinating to improve vaccine acceptance. Our study findings revealed that unique pandemic-related factors play an important role in COVID-19 vaccine acceptance, including vaccinating for emotional health, resuming social activities, and vaccine mandates. Parents and adolescents expressed social reasons for wanting to vaccinate, such as resuming in-person activities after having experienced social isolation during the pandemic.

As with other adolescent vaccines such as HPV vaccination, having health care practitioners take a nonconfrontational announcement approach to explicitly recommend COVID-19 vaccination (as the default and recommended action rather than a question to parents) may be needed to more effectively reach parents who either mistrust the vaccine or view COVID-19 risk as manageable [39,42]. Parents may benefit from being reminded of the rapidly changing risk environment, including the relaxation of restrictions, delta and omicron variants, elimination of mask mandates in many states, and inconsistent protective behaviors of others. Vaccination can prevent severe COVID-19 disease, given that adolescents no longer live in protected “bubbles”.

Parents also expressed vaccinating their children in response to policy cues to prioritize vaccination through mandates, whether for school, travel, state, or work mandates. Mandating vaccination signals prioritizing the recommended behavior in response to societal risk [21]. Social, emotional, and policy options are important to consider for pediatric, school nurse, and public health official messaging. Listening to parents’ concerns and questions and answering their questions continue to be important as a point-of-care educational opportunity to allay parents’ concerns and build trust [43].

Study limitations include the fact that findings reflect parental perspectives prior to the emergency authorization of COVID-19 vaccination for adolescents, during a rapidly evolving risk environment. At the time of this study, no vaccines had been approved for adolescents, and this discussion was one of possibility. Adults who were health care workers and more recently, teachers, had received the opportunity to be vaccinated, but there was no timeline for when adolescents would have approval and access to vaccines. The identified factors continue to be critical for effective messaging to vaccinate the remaining 47% of unvaccinated adolescents in the United States, who are at 10-fold higher risk for potentially more severe outcomes should they contract COVID-19. In contrast to the United States, countries such as the United Kingdom have not prioritized vaccinating adolescents as a policy until recently [8,9,44].

This study explored perspectives of parents and adolescents in Orange County, California, and reflects a time when most adolescents there were being schooled online from home. Orange County’s demographics reflect a diverse racial and ethnic makeup of predominantly White, 35% Hispanic (Mexican and El Salvadorean), and 22% Asian (e.g., Vietnamese, Korean, Chinese, and Indian) [18], distinct from not only the rest of the country but also neighboring rural counties, such as Imperial County, in Southern California. Parents’ vaccine attitudes and access may vary significantly in neighboring rural Imperial or urban Los Angeles counties. Interviews with parents from different regions of the country such as in the Southeast or who have low educational attainment may reveal additional factors such as vaccine access (finding time to take off work to vaccinate adolescents) related to vaccine attitudes. School-based vaccine clinics may benefit families in such cases. Our recruitment strategy sampled parents who mirrored the community in which the study was conducted, with diversity in race and ethnicity, multigenerational households, children with chronic conditions, and children who were schooled online. A notable limitation is that the education level of parent participants was relatively high, with almost three quarters having a college education. This may have obscured other concerns among families with lower levels of educational attainment. We know from other studies that low-income communities have expressed COVID-19 vaccine uncertainty [21,45]. A Kaiser Family Foundation study revealed that low-income parents are more concerned about taking time off work and traveling to a vaccine site to vaccinate their children [46]. Thus, vaccine constraints—one of the 5Cs not found in this study—could play a role for some parents not sampled in this study. Another limitation is that we did not collect data on religiosity, which the literature has shown can influence vaccine attitudes in some cases. Finally, the limitation of conducting virtual (Zoom) parent discussions must be recognized. The disadvantages of virtual discussion groups include not being able to read nonverbal cues as accurately, technology glitches, not being able to control for others present or listening on the call (i.e., lack of privacy possibly affecting what a person discloses), and the sample being limited to those who have Zoom technology. On the other hand, advantages of virtual discussion groups include cost and time savings, effective access and convenience, expanding the geographic range of inclusion, and improving the inclusion of parents who might otherwise not be able to attend.

## 5. Conclusions

With the Pfizer BioNTech vaccine authorized for use in children aged 5 or older in the United States, public health messaging, especially by pediatricians, school administrators, nurses, pharmacists, local community leaders, and public health officials, will continue to be paramount in addressing the educational needs of parents and adolescents to bolster vaccine confidence and the understanding of how vaccines protect and benefit children’s health [38,47,48]. Continued parent education about vaccines is needed, including eliciting and addressing questions from parents and adolescents. Educational opportunities for parents and adolescents may be more accessible, either through schools or local community leaders, to answer parents’ questions as part of trust building [45,48,49]. In discussions with families, pediatricians, community, and school clinic workers should walk the tightrope of being transparent about possible rare vaccine side effects such as myocarditis [26] or anxiety-induced syncope [50] while allaying parents’ and adolescents’ vaccine concerns. Parent vaccine messaging may benefit from emphasizing that through vaccination, their children will be more protected and can resume social activities, sports, travel, and in-person school or summer camps safely, with less likelihood of acquiring and transmitting COVID-19 and less disruption to their adolescents’ lives. Messaging to reach hesitant parents may also include messaging via adolescents, who in some cases have greater and independent access to COVID-19 vaccine information and a greater willingness to vaccinate than their parents.

## Figures and Tables

**Figure 1 ijerph-19-04212-f001:**
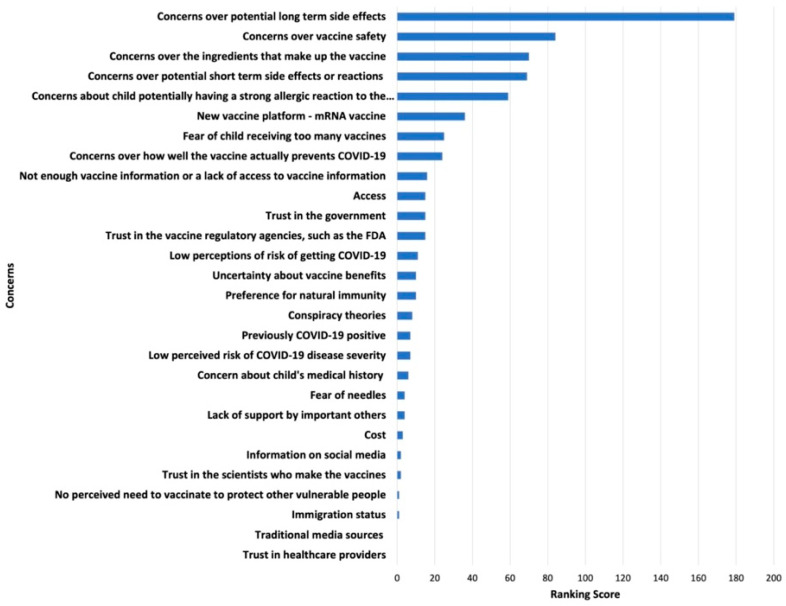
Top Parental Concerns for Vaccinating Adolescents against COVID-19.

**Figure 2 ijerph-19-04212-f002:**
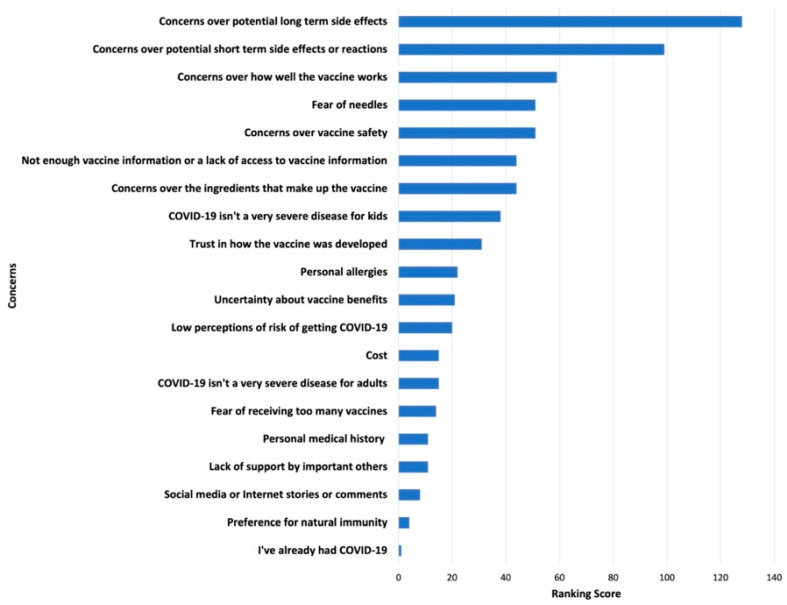
Top Adolescent Concerns for Vaccinating against COVID-19.

**Figure 3 ijerph-19-04212-f003:**
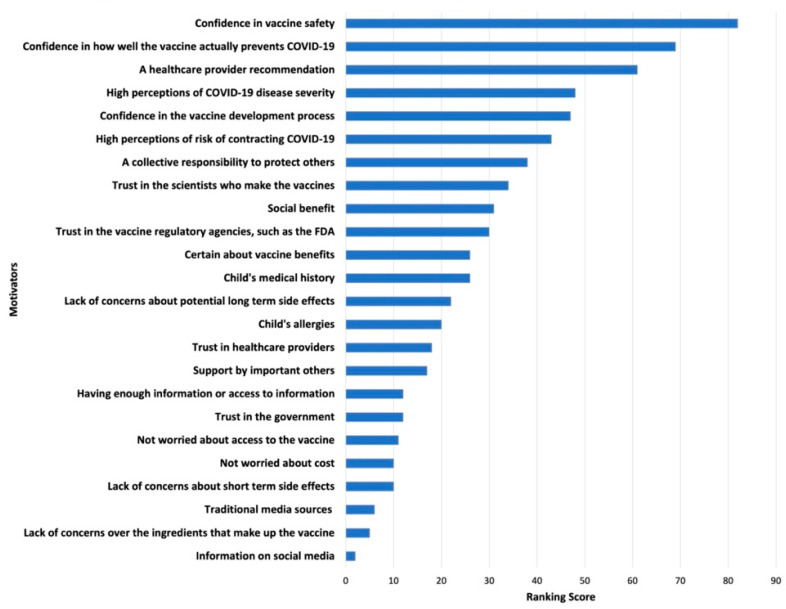
Top Parental Motivators for Vaccinating Adolescents against COVID-19.

**Figure 4 ijerph-19-04212-f004:**
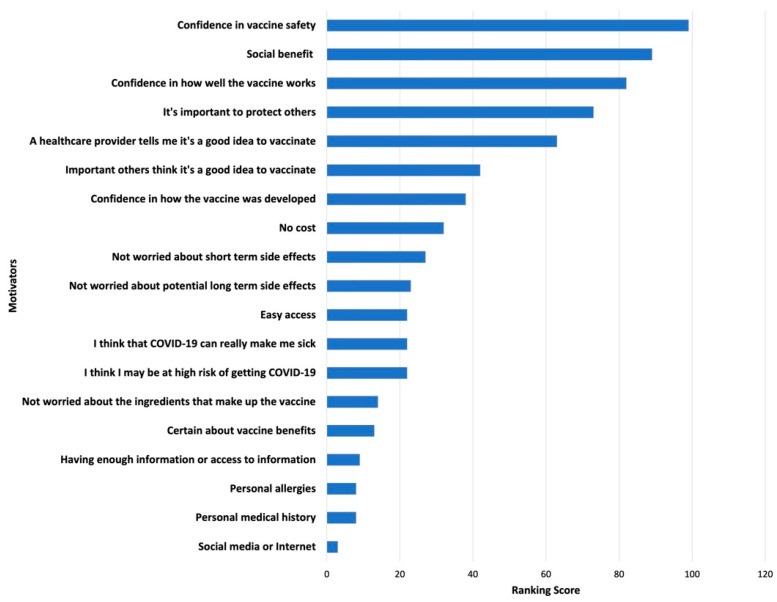
Top Adolescent Motivators for Vaccinating against COVID-19.

**Figure 5 ijerph-19-04212-f005:**
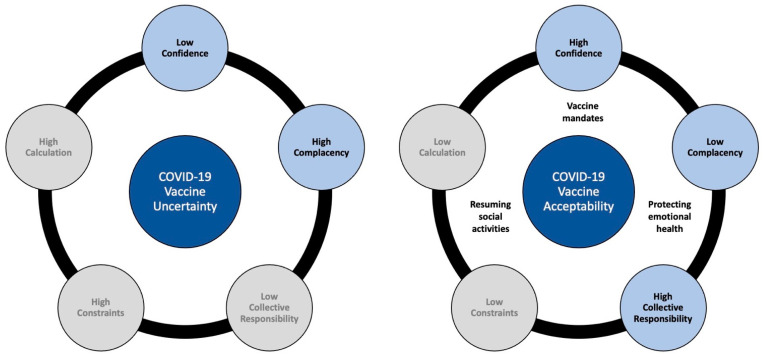
Graphical Representation of Qualitative Results.

**Table 1 ijerph-19-04212-t001:** Overview of Survey Question Domains for Parents.

Topic	Example
Parent–Child Communication	I have asked my adolescent child about their thoughts and opinions regarding the COVID-19 vaccine. (5-point Likert scale)
Ranking Vaccine Concerns and Motivators (5C Vaccine Hesitancy Model)	When thinking about vaccinating your adolescent, please rank your top five COVID-19 vaccine concerns…
Confidence	Concerns over potential long-term side effects. Trust in health care providers.
Complacency	Low perceived risk of getting COVID-19. Low perceived risk of COVID-19 disease severity.
Constraints	Cost, access.
Collective Responsibility	No perceived need to vaccinate to protect vulnerable people.
Calculation	Uncertainty about vaccine benefits.
Parenting Style	In general, how are the most important health decisions made between you and your adolescent child?
Positive Expectancies	I think the COVID-19 vaccine will protect my adolescent child from the COVID-19 virus. (5-point Likert scale)
Vaccinating Under Conditions of Uncertainty	From information that I have been able to find, I think the COVID-19 vaccine is safe for my adolescent child. (5-point Likert scale)
Information Sources	Indicate how much you trust vaccine information from medical professionals. For example, doctors. (5-point Likert scale)

**Table 2 ijerph-19-04212-t002:** Focus Group Discussion Guide.

Topic	Example
General Vaccine Attitudes and Warm Up	Tell me about how you made the decision about whether or not to vaccinate your adolescent the last time you were asked to do so?
Parent COVID-19 Vaccine Decision Making	5C Vaccine Hesitancy Model.
Confidence	Describe to me what you know about COVID-19 vaccine safety and effectiveness in adults and children.
Complacency	Describe your perspective on whether your adolescent needs the COVID-19 vaccine and whether you think your child is at risk for getting COVID-19.
Constraints	Tell me how confident you feel that you will be able to get your children vaccinated once it becomes available?
	Tell me about any potential barriers that might impact you (e.g., insurance, cost, finding out where and when you can vaccinate your child, availability).
Collective Responsibility	Tell me how helpful the vaccine will be for you, your child, your household or your community (e.g., herd immunity).
Calculation	Explain to me what you plan to do about vaccinating your adolescent once the vaccine becomes available and recommended?
	Tell me about the things that you consider in your decision (e.g., the health of your child, the opinions of others, preference for natural immunity, etc.).
Vaccine Acceptability After Emergency Use Authorization	What information or actions, if any, would make getting the vaccine more acceptable?
COVID-19 Vaccine Information Sources	Where or who do you turn to for information about the COVID-19 vaccine?
Vaccinating Children Younger than 12 Years (if applicable)	For those of you who have children under the age of 12 in the home, how are your decisions about whether or not to have them take the vaccine the same or different?

**Table 3 ijerph-19-04212-t003:** Parent and Adolescent Dyad Demographics (*N* = 46).

Demographics	*n*	% or M (SD) ^1^
Parent age (in years)	-	44.5 (6.1)
Adolescent age (in years)	-	14.1 (1.7)
Parent gender		
Female	45	98
Male	1	2
Adolescent gender		
Female	19	41
Male	26	57
Transgender, male	1	2
Parent race and ethnicity		
Latino	22	48
Caucasian	13	28
Asian	9	20
American Indian	1	2
Pacific Islander	1	2
Parent nativity status		
U.S. born	19	41
Foreign born	27	59
Parent education		
Less than high school	1	2
High school	3	7
Vocational or technical school	1	2
Some college	3	7
Associate degree	4	9
Bachelor’s degree	11	24
Graduate school	23	50
Parent occupation		
Business	13	28
Education	11	24
Health care	4	9
Social services	3	7
Other	15	33
Multigenerational household	13	28
Adolescent school type		
Public	36	78
Charter	6	13
Private	4	9
Adolescent schooling delivery mode		
Online distance learning	30	65
In-person or hybrid	16	35
Adolescent chronic conditions	10	22
Asthma	4	9
Asperger’s	1	2
Mental illness	2	4
Epilepsy	1	2
Obesity	1	2
Seasonal allergies	1	2
Adolescent influenza vaccination status		
Vaccinated in the past year	31	67
Not vaccinated in the past year	15	33

^1^ SD = standard deviation.

**Table 4 ijerph-19-04212-t004:** Uncertainty in Vaccine Decision-Making.

5C Model Construct	Construct Details	Participant Quotes
Low confidence in the COVID-19 vaccine	Vaccine safety concerns: long-term side effects, exacerbating existing chronic conditions, vaccine ingredients, and more studies needed	“My concerns are just what the long-term side effects are and how each individual would react to it”—Parent 11
		“My daughter, like I said, the youngest one does have underlying issues and she already has things going on with her, so I don’t want to cause any more on her”—Parent 32
		“Because like I said, the ingredient, we don’t know what’s in it, so we don’t know how it’s gonna affect our children”—Parent 17
		“Well, for me personally it’s just a matter of time, and I want to see more testing—basically more people, just more studies being done, and of course, have more research articles coming out on the vaccine overall”—Parent 3
	Vaccine efficacy concerns	“The other thing that I wanted to mention was with all the variants coming through, you know, and so that was the other concern. Is this going to be effective, still be as effective with new variants?”—Parent 20
	Trust concerns: mistrust in pharmaceutical companies, regulatory agencies, public health authorities, and health care providers	“This vaccine was developed super-fast. So, you know, I don’t have a lot of trust about that”—Parent 25
		“If you’re on the J&J [Johnson & Johnson] panel, everyone’s expecting it [the vaccine] to be rolled out and in arms like next week. Who’s gonna be the one that’s like, ‘Well, I think that there might be a concern,’ so that does concern me a little bit because whistleblowing is hard enough”—Parent 15
		“There is a lot of business and tremendous politics involved in it, and that makes a common person worrisome—like OK, is the government or are the authorities really concerned about the common person’s health or is there big business involved in it and that’s why there is a lot of promotion to get the vaccine all of a sudden?”—Parent 19
		“It’s not just about public health at all or the good of the people. There’s just so many other factors involved, which, you know, unfortunately, in the area of public health, it shouldn’t be, but it is”—Parent 21
		“They [CDC] did not give us the accurate information to begin with. They said mask is not needed, but it was needed from the first day. So, I’m thinking about vaccine. They may say, no, the vaccine is not needed for younger children and when everybody goes back to normal, we may have a peak for people that are not vaccinated yet. That’s my concern”—Parent 13
		“Pfizer and Moderna had been kind of secretive; what’s in that vaccine is not really public knowledge, so that makes me suspicious. Why is it not common knowledge?”—Parent 3
High complacency regarding COVID-19 disease risk	Low perceived risk of contracting SARS-CoV-2	“My kids are still in distance learning … and so, I don’t think they’re at risk at this time”—Parent 36
		“She’ll have a higher than usual chance of getting it at college versus high school, middle school where they’re coming home, and you have some control of what they’re doing socially, so like right now, I don’t think she’s getting COVID any time before she leaves home”—Parent 15
		“As a family, we do everything wrong. We host parties. My son has a huge party. We had a huge party. We went to gather with family for the holiday. We travel. I purposely tried to get COVID, and I’m not getting it! Then I have people like my neighbor who are always home, and they have COVID. So, I just find it mind blowing, like, we are trying to get COVID! No masks, everywhere touching everything, especially my youngest with everything in his mouth, and we can’t get it, you know? [laughs] So, it’s like why the heck do we even need a vaccine?”—Parent 3
		“I don’t know if they need it. We were sick and they tested negative, all of the kids. I feel like for kids it’s a little bit … like their immune system. I don’t know if it’s—but all adults here got it and none of the kids did”—Parent 40
	COVID-19 perceived as a mild disease	“They are fine with COVID because my son, he doesn’t have, like, any chronic disease or no allergies, so I think he’s fine if he has it”—Parent 16
		“Like I said, we don’t get sick very often, so I think their immune systems are pretty strong and that if they were to get it, they would have a good outcome”—Parent 37
		“We’ve seen it, I guess, with a lot of adolescents and they’re all fine—fine meaning it seems like it’s not even as bad as the flu”—Parent 15
		“Like, it is more of an inconvenience to be really honest with you, right, you know, it’s more of a like scarlet letter on your chest, right, you know what I mean, and an inconvenience in your life”—Parent 20
		“The COVID actually [is] not very harmful for the kid under 16, so for the COVID vaccine, I think like I said, I will wait maybe for one or more years”—Parent 17
		“You know, she is very young. Is it better just to get COVID, get it over with? And who knows if those antibodies last either, but they don’t, similarly, they don’t know what the vaccination is going to do. I don’t know, so we’re on the fence”—Parent 15

**Table 5 ijerph-19-04212-t005:** Vaccine Acceptability.

Relevant 5C Model Construct	Construct Details	Participant Quotes
High confidence in the COVID-19 vaccine	Trust in health care providers; increasing confidence because an increasing number of others have vaccinated without complications; family norms to vaccinate	“I go with the recommendation, you know? As soon as it’s ready and available for our children, we’d certainly do it. My husband and I already agreed we’d all go do it”—Parent 14
		“I rely on the, you know, the expert advice of two people that I trust and personally, that’s my doctor and my children’s doctor”—Parent 2
		“Since I can remember, my parents have always taught me that vaccines are good. I’m educated as far as health. Regardless of what is said, whether they’re good or bad, the things I have learned since childhood is that vaccines are good”—Parent 23
Low complacency regarding COVID-19 disease risk	Protecting the child’s health	“I’m more concerned actually about the virus”—Parent 4
		“Once my 14-year-old can get the vaccine, I’m going to feel even more relief, because he does have, you know, a lung issue, and I worry for him a lot because I’ve watched him over the years as we’ve had nebulizer treatments and, you know, there’s that cough that he gets, that’s so scary for us”—Parent 35
		“I want my children alive. I want them to have a future, and I want to be alive”—Parent 41
		“[Vaccinating] is healthy because they get sick less. Back then when they were going to school, you know that they’re sick it’s so uncomfortable even for you as an adult, to be at work or on the street, sneezing, blowing your nose. So, it prevents you from getting sick. That’s why we all decided to get immunized”—Parent 23
High collective responsibility to vaccinate	Vaccinating for one’s social circle	“I would like them to have the vaccine, because we have to care about teachers, you know, school staff and others, too. It’s not only family members, it’s neighbors, you know, everyone, grocery staff everywhere”—Parent 12
		“I want my children to be protected and also, in a sense, you know, they’re protected and then they’re also protecting the people around them. So, I feel like it’s so critical, and I would like them to get vaccinated”—Parent 26
		“I would say he needs to be vaccinated. We also have, you know, an older parent living with us. … Some people get really sick, and it never seems to be something that is really predictable”—Parent 33
		“Yeah, that’s my feeling about my 14-year-old is that just how he could potentially spread it in the community, you know—if like over the summer he’s involved in sports and he’s with a group of kids, he gets it, and then, you know, he’s not wearing his mask and, you know, inadvertently spreads it to, you know, somebody who’s not able to fight it off or whatever or wasn’t able to get the vaccine for whatever reason, and you know, and if something ever happened to somebody else as a result of my son, that would be really difficult”—Parent 35

**Table 6 ijerph-19-04212-t006:** Pandemic-Related Factors in COVID-19 Vaccine Decision-Making.

Theme	Participant Quotes
Protecting child’s and adult’s emotional health	“My son plays sports and so far, we do take him for his games, but there’s always a constant fear. What if he contracts it, right? He may not get impacted, but if he’s a carrier and we get it, then what happens, right? So yeah, that might be one benefit of getting your child vaccinated—that you don’t have to live in a constant fear”—Parent 16
	“At this point, mental health, not so much for the adults, but for children and teens in particular. Yeah, it’s been a challenge”—Parent 44
	“I think mentally, too, it gives a sense of security, right, that we’re all protected to some extent”—Parent 46
	“I think it [vaccinating] would be very helpful, has been like a sense of relief. At least just a little bit, you know, lowering that risk of exposure to my children and into the community because we, you know, like a lot of us, we don’t go out, but we go to church or we go to the supermarket and just want to do what I can at least on my part to … be safe”—Parent 32
Resuming social activities	“I feel bad for my senior. He lost half of 11th grade and now he’s a senior and applying for colleges and I feel like he’s going to still be a senior going into college. … I just feel so bad, you know. And my daughter, who graduated eighth grade, didn’t finish eighth grade, didn’t have her ceremony, and then now she’s a freshman and has not stepped foot on high school campus—has no idea. So, it’s almost like all of our kids are one grade behind mentally. … It’s like they lost an age”—Parent 26
	“It’s just the fact that once the child gets vaccinated, they can engage in more activities”—Parent 16
	“My husband’s family, there are kind of a lot of seniors in the family, and they all got vaccinated. … They said if you don’t get the vaccine, you’re not welcome to the party”—Parent 12
	“I don’t think I’m going to be the first one to line up for it, but I also want to be able to see people right, you know what I mean, and do certain things, so if that helps me or helps people be able to feel more comfortable, then I would consider it more”—Parent 20
Vaccine mandates	“I guess if it was mandated, for instance, to attend whether it’s high school or college or to fly, then obviously, you know, we’re going to get it”—Parent 15
	“It might be more related to things we value, rather than information. So, work opportunities or doing things that are important to us like being able to travel. We have family who are spread out both internationally and across the country, so if it was required for travel, we would because that that would supersede our perceived risk or hesitation to take it”—Parent 37

## Data Availability

Not applicable.

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
