# Peer review of "Adolescent COVID-19 Vaccine Decision-Making among Parents in Southern California"

_ijerph, 2022, doi:10.3390/ijerph19074212_

Round 1
Reviewer 1 Report
Reviewer comments are attached.

Reviewer 2 Report
Ever since the Pfizer BioNTech vaccine for use in children aged five or older was authorized a heated debate over the usefulness of coronavirus vaccines for children emerged. Thus, the unquestionable strength of the manuscript is that its topic is very important and timely and the research itself was designed and described clearly. I also appreciate the fact that the study was guided by the5C vaccine hesitancy model. Moreover, while there are many research on the vaccine hesitancy in numerous countries still there is a scarcity of previous work on parent-adolescent COVID-19 vaccine decision-making. Thus, I believe that the research fills the gap in the literature and may be of interest to the readers of the Journal.
Another advantage of this study is the qualitative approach which enabled the Authors to identify and understand the ways in which parents expressed COVID-19 vaccine acceptance or uncertainty for their adolescent children. This in turn, provided in-depth knowledge about parents’ hopes and anxieties related to COVID-19 vaccine. This is of special importance because identifying factors associated with vaccine acceptance and uncertainty may inform effective vaccine messaging. Thus, while some policymakers, scientists and the media tend to stigmatize parents’ uncertainty and/or hesitation for vaccination of their adolescent children as ‘irrational’, ‘unscientific’ and/or ‘anti-vaccination’ the Authors have managed to highlight the beliefs and ideas that underline parents vaccine decision-making. Finally, while describing already known factors associated with vaccine acceptance/hesitancy the research describes three additional factors that were not recognized in vaccine specific theories and that are unique motivators to the pandemic and may motivate parents to vaccinate their children. This in turn, may help to build more effective vaccination campaigns, especially as we face another wave of the COVID-19 pandemic.
The only concern I have is that I am a bit surprised that apart from many parents’ and adolescent’s demographics no information on such important factors as religion/spirituality and self-perceived status health was given. Meanwhile, research show that they also influence individuals’ attitudes towards vaccination.
However, apart this small remark I appreciate this research a lot. Thus, while I wish to congratulate the Authors their interesting research I recommend its publication. I am convinced that the issues raised in the article will help to understand beliefs and arguments that underline parents vaccine decision-making and will stimulate the discussion on the effective vaccine messaging by healthcare practitioners, public health and school officials.
Author Response
We thank reviewer 2 for their comments. We agree and have noted R2’s comment about not having collected data on the religious/spirituality affiliation and perceived health status of participants (we included that in our study on vaccinating younger children). We thank the reviewer for this observation. We agree that this would be informative data and aim to collect such information for a future study. We add to the limitations that religious affiliation was not collected.
Reviewer 3 Report
The manuscript investigates the underlying factors behind vaccine hesitancy for adolescents. In particular, the study is focused on Southern California households from socioeconomic groups known for higher covid-19 mortality and morbidity risk. This is an important and timely issue, given the relatively low vaccination rates among adolescents in the U.S.
This study is based on a combination of surveys and focus groups for 46 parent-adolescent dyads. The analysis and survey methodology are only partially described. Neither the survey nor the focus groups scripts are available. The results from the survey are only partially available (descriptive statistics of the sample) and no quantitative evidence on the focus groups analysis is provided.
The study may have been conducted an analyzed rigorously and the conclusions may be supported by the results, but this reviewer could not assess it with the information provided.
Main issues
The survey fielded is not provided and its results are only partially described.
Beyond questions related to the socioeconomic profile of the participants, no information regarding the results of the survey is provided. According to the authors, “Survey domains included questions regarding: general and 119 COVID-19 vaccine attitudes, COVID-19 vaccine intent for adolescents, parenting styles, positive vaccine expectancies, vaccine decision-making under conditions of uncertainty, vaccine information sources, and the ranking of concerns and motivators related to vaccinating adolescents against COVID-19”. Why aren’t any of the responses to the survey reported in the present study? In addition, survey questions are not available to the reviewers.
The findings are almost exclusively based on the focus groups analysis, but the study’s claims are not backed by supporting evidence.
Almost the entirety of the analysis in the paper is devoted to the content of the focus groups. Yet, almost no evidence, descriptive or otherwise, is provided on the content of the discussions. According to the authors, a team of coders classified the themes in the focus group contents, but no quantitative evidence is provided of the results.
Focus group discussions contents are always described in a vague manner. Some examples:
“A few participants doubted public health agencies such as the CDC given their inconsistent mask wearing recommendations.” (line 191)
“Many parents did not perceive COVID-19 as a significant health threat to their adolescent.” (line 204)
What does few or many mean in this context? Without any supporting evidence, the descriptions are vague and difficult to validate for the reader. The quotes from the focus groups, while interesting, do not constitute sufficient evidence on their own.
This issue extends to the analysis of the reported associations. For example, the authors claim that “Greater confidence in the COVID-19 vaccine was also associated with personally knowing people who had vaccinated without side effects or complications.” (line 244). How has this association been established? The issue is not limited to this specific example, but extends to all the reported associations.
Minor issues
Recruitment and sample construction procedures are not sufficiently clear.
How many dyads applied and how many were turned down? Why were they turned down? Was there any attrition (participation in survey but not the focus group)?
Conclusion
This research studies the determinants of vaccine hesitancy for adolescents. While the topic is timely and important, in my opinion the methodology is not sufficiently transparent, and the claims are not appropriately backed by evidence. For these reasons, I believe that this manuscript is not publishable in its present form.
Reviewer 4 Report
Dear Authors,
some methodological aspects should be explained and described:
- Why was the number of participants in each focus so low? Usually, a focus group involves 6-10 people.
- Additionally, the authors should make explicit the criteria by which they decided which focus groups to conduct, i.e. by which criteria survey participants were assigned to the various focus groups.
- Authors should state how they determined the sample size -number of focus groups; this is important not only in quantitative analysis but also in qualitative analysis.
Finally, regarding the results, it would be helpful to the reader, if the results of the qualitative analysis were graphically illustrated through mind maps.
Author Response
Dear Authors,
some methodological aspects should be explained and described:
- Why was the number of participants in each focus so low? Usually, a focus group involves 6-10 people.
Response. We thank the reviewer and clarify. First, we would like to clarify and remind reviewers that these focus groups were conducted during pandemic lockdown and were virtual. As a result, there were shorter focus group discussion time (60 rather than 90 minutes for parents; 30 minutes for adolescents) and additionally, we had parent-adolescent dyads on the virtual zoom focus group discussion, which involved more people and we wanted to ensure that each participant contributed in a meaningful way to the discussion. Therefore, 4-6 parent-adolescent dyads was a more appropriate number to allow for meaningful contributions and conversation. We did have a range of focus group size from 3-6. Prior to the pandemic, we would typically have focus groups that were in-person and 90 minutes, but given the pandemic circumstances, one-hour virtual focus groups we found that 8-10 people in a one-hour virtual focus group does not allow adequate time for all members to contribute to discussion. We aimed for having 4-5 parents and adolescents on the FG, but depending on scheduling since we wanted to hear from as many parents as possible given the February/March 2021 time period prior to the EAU (emergency authorization) of COVID-19 vaccination for adolescents (which occurred May 2021) with early evening scheduling it ranged from 3-6 families participating at a time.
- Additionally, the authors should make explicit the criteria by which they decided which focus groups to conduct, i.e. by which criteria survey participants were assigned to the various focus groups.
- Response. Parents were assigned to focus group dependent on criteria of eligibility namely, living in Orange County, CA, having adolescent children. Spanish speaking only parents were assigned to FG that were moderated and held in Spanish.
- Authors should state how they determined the sample size -number of focus groups; this is important not only in quantitative analysis but also in qualitative analysis.
- Response. Per Hennink et al (2019) [What influences saturation? Estimating sample size in focus group research] we had a total of 12 focus groups with 9 in English and 3 in Spanish. Within these focus groups, to capture the greatest possible range of vaccine attitudes we also sampled parents by occupation, race/ethnicity, whether child has a chronic medical condition, and whether family lived in a multi-generational household. Based on Hennink et al, 2019, focus groups stratified by a characteristic (Spanish speaking parents) requires 2-3 focus groups to identify the majority of issues across data. Additional focus groups may be needed to identify meaning saturation clustered with one-two additional focus group included per characteristic (since we also examined race/ethnicity, multi-generational household, child with chronic condition, parent occupation) we had 9 in English. We examined codes and themes within and across groups, with meaning saturation occurring across the data.
Finally, regarding the results, it would be helpful to the reader, if the results of the qualitative analysis were graphically illustrated through mind maps.
Response. We provide a graphic mind map of results in addition to the graphics of ranking of concerns and motivators.
Reviewer 5 Report
This is an interesting and timely manuscript reporting on a qualitative study of parent/adolescent dyad decision making regarding the COVID-19 vaccine for adolescents.
Overall, I feel that the manuscript is decently presented and methodology reasonably described. Some editing for writing quality/wording is needed.
A few minor notes:
- The focus groups seem quite small given that the standard focus group in social science research is typically 8-10 participants. Were the groups purposefully set up to be this small?
- Can the reader assume that only the people who indicated an unwillingness to get their adolescent vaccinated are represented in Table 4 and only those who were willing to do so are in Table 5? This isn't entirely clear.
- Line 377 should refer to Table 6, not 4
Author Response
This is an interesting and timely manuscript reporting on a qualitative study of parent/adolescent dyad decision making regarding the COVID-19 vaccine for adolescents.
Overall, I feel that the manuscript is decently presented and methodology reasonably described. Some editing for writing quality/wording is needed.
A few minor notes:
- The focus groups seem quite small given that the standard focus group in social science research is typically 8-10 participants. Were the groups purposefully set up to be this small?
Response. We clarify that given these were virtual focus groups (during the pandemic lock down) and included parent-adolescent dyads hence, additional people per unit family.
In the evenings with parents and their adolescent child, conducting a one hour discussion with parents (after a half hour discussion with the adolescents), having 4-5 parent-adolescent dyads on the virtual discussion group was the appropriate number of parents in order to have each parent and separately, adolescents contribute meaningfully to discussions. We report in the methods section that we had a range from 3-6 parents per group. Based on prior experience and given the virtual discussions, more than 6 would become problematic for having all contribute to discussion.
- Can the reader assume that only the people who indicated an unwillingness to get their adolescent vaccinated are represented in Table 4 and only those who were willing to do so are in Table 5? This isn't entirely clear.
- Response. Survey responses indicated that 46% were willing to vaccinate, 11% were not, and 43% were unsure. Given that vaccinating adolescents was still a hypothetical scenario, there were no news reports at the time about when the adolescent COVID-19 vaccine would become available under emergency authorization, we analyzed all data comprehensively across all focus groups. We did not analyze first within groups’ intent to vaccinate and then across given the hypothetical scenario at the time of the focus groups and that vaccination for parents had just become available but only for some (healthcare workers and teachers). We’ve conducted studies prior where parents are hesitant even though they had vaccinated their children.
- Line 377 should refer to Table 6, not 4
- Response. Thank you. We have corrected this.
Round 2
Reviewer 3 Report
I thank the authors for their thoughtful responses. The updated version of the manuscript addresses several of the initial concerns of this reviewer (e.g. it provides the surveys and guidelines for the focus groups). There remains, however, a fundamental disagreement on the analysis of the contents of the focus groups.
The authors provide a thorough argumentation on the approach taken in the manuscript, including supporting references. In a nutshell, instead of reporting quantitative evidence, they rely on “semi-quantitative” descriptors (e.g. a few, many , several). This “semi-quantification” is based on the analysis of the videos of the focus groups. While the evidence used to decide whether a point was made by “a few” or “several” parents was based on the (coded) frequencies of the different themes, the authors choose not to include these or report the underlying decision rules (e.g. between 1-5 parents constitutes a few).
This is, in my opinion, problematic. The reported associations are ultimately based on subjective assessments that hidden to the reader. One may or may not agree with the rules used by the authors to report their findings. The same can be said about the reported associations in the manuscript. This constitutes a disagreement on the validity of the methodological approach that is unlikely to be resolved with further revisions. For these reasons, I believe that this manuscript is not publishable.
Author Response
We thank reviewer 3 (R3) for their time reviewing. We have made efforts to be transparent about the data analysis process. Discovering themes relevant to COVID-19 parent vaccine decision-making are not correlated with frequency of codes necessarily, but we provide frequency of codes as one aspect of data analysis transparency. We describe in detail in the data analysis section of Methods, our data analysis. This involves 3 steps: data immersion (reading and becoming familiar with data); primary level data analysis which involves tagging segments of data with descriptive codes; secondary level data analysis which involves organizing existing codes into themes, involves interpretive data analysis (which yes is subjective as all qualitative data analysis are; but we are transparent about the process of our analysis); at this level of analysis we revisit our research questions and also deductively, revisit how parent conversations about vaccinating their children (or not) [again, this was all in the hypothetical as the COVID-19 vaccine was not yet authorized for adolescents] and how parent narratives reflect aspects of the 5C vaccine acceptance model (confidence, complacency, constraints, collective responsibility, calculation) that shaped uncertainty or acceptance. We then identified other, additional pandemic unique factors. This was in the data and the identification of additional factors is not necessarily correlated with frequencies (even low prevalence codes can contribute to building a comprehensive understanding of a phenomenon) but rather whether new themes emerged, were expressed, and were qualitatively distinct.
Reviewer 4 Report
None